# Co-ERA-Net: Co-Supervision and Enhanced Region Attention for Accurate Segmentation in COVID-19 Chest Infection Images

**DOI:** 10.3390/bioengineering10080928

**Published:** 2023-08-04

**Authors:** Zebang He, Alex Ngai Nick Wong, Jung Sun Yoo

**Affiliations:** Department of Health Technology and Informatics, The Hong Kong Polytechnic University, Kowloon, Hong Kong SAR, China; zebang.he@connect.polyu.hk (Z.H.); nn-alex.wong@polyu.edu.hk (A.N.N.W.)

**Keywords:** co-supervision, COVID-19 chest infection segmentation, enhanced region attention

## Abstract

Accurate segmentation of infected lesions in chest images remains a challenging task due to the lack of utilization of lung region information, which could serve as a strong location hint for infection. In this paper, we propose a novel segmentation network Co-ERA-Net for infections in chest images that leverages lung region information by enhancing supervised information and fusing multi-scale lung region and infection information at different levels. To achieve this, we introduce a Co-supervision scheme incorporating lung region information to guide the network to accurately locate infections within the lung region. Furthermore, we design an Enhanced Region Attention Module (ERAM) to highlight regions with a high probability of infection by incorporating infection information into the lung region information. The effectiveness of the proposed scheme is demonstrated using COVID-19 CT and X-ray datasets, with the results showing that the proposed schemes and modules are promising. Based on the baseline, the Co-supervision scheme, when integrated with lung region information, improves the Dice coefficient by 7.41% and 2.22%, and the IoU by 8.20% and 3.00% in CT and X-ray datasets respectively. Moreover, when this scheme is combined with the Enhanced Region Attention Module, the Dice coefficient sees further improvement of 14.24% and 2.97%, with the IoU increasing by 28.64% and 4.49% for the same datasets. In comparison with existing approaches across various datasets, our proposed method achieves better segmentation performance in all main metrics and exhibits the best generalization and comprehensive performance.

## 1. Introduction

Chest infections, a medical condition resulting from the invasion and proliferation of microorganisms such as bacteria, viruses, or fungi within the chest area, particularly the lungs, present various symptoms like coughing, chest pain, shortness of breath, fever, and fatigue. Besides physical symptoms, radiology imaging tests, including CT and X-ray scans, can detect chest infections. Radiology techniques including CT imaging and X-ray imaging allow radiologists to visualize infection extent and location in the chest cavity, significantly aiding in accurate diagnosis and effective treatment planning. In pandemics involving chest diseases, such as the previous COVID-19 outbreak, a reliable and efficient diagnostic method for identifying infection regions in chest radiology scans is crucial for monitoring disease progression and devising appropriate treatment strategies. However, radiologist shortages can hinder accurate chest infection diagnosis and impede infection region identification efficiency. Consequently, an automated tool for delineating infection regions from chest radiology scans is vital for facilitating chest infection diagnosis.

Advancing deep learning algorithms offer the increasing potential for automating infection region segmentation in radiology scans, including CT or X-ray, presenting an opportunity to reduce the manpower and time required for radiologists in infection region identification. However, existing deep-learning models primarily focus on analyzing entire radiology images, rather than specifically targeting lung regions where infection signs are more likely to appear, possibly leading to segmentation of areas outside the lung region and compromising diagnostic accuracy, which is shown in Figure 1.

To enhance the precision of infection segmentation in these radiology scans, we postulate that infections manifest within the lung region. Building on this assumption, there are two potential solutions to enhance segmentation performance: the first method entails using ground-truth lung region masks to isolate the lung region before segmenting the infection region, while the second method involves employing a separate lung segmentation model to identify the lung region prior to infection segmentation. However, both approaches have drawbacks. The first method is constrained by the potential absence of ground-truth lung region masks in new cases, while the second method entails a high time cost due to sequential model usage and lacks the ability to share features between lung and infection segmentation models. As a result, a novel network must be developed to overcome these limitations, enabling more efficient and accurate infection segmentation in radiology scans by better leveraging lung region information.

In the paper, we present the Co-ERA-Net, a new deep-learning model for accurately segmenting infection regions in Chest images while overcoming the limitations of existing algorithms and direct solutions. Our proposed Co-ERA-Net involves parallel flows for both the lung region and infection region, where co-supervision is achieved by both lung region information and infection region information. Instead of relying on sequential segmentation of the lung and infection regions, our Co-ERA-Net uses Enhanced Region Attention Module (ERAM) to connect the lung region flows and infection region flow by integrating the information between these flows and highlighting the regions with a high probability of infection. In this way, the Co-ERA-Net can minimize segmentation areas outside lung regions and boost segmentation performance.

We conducted a series of experiments to evaluate the efficacy of the proposed Co-ERA-Net, including comparing our network with various state-of-the-art models, performing an ablation study to investigate the contribution of the co-supervision and enhanced region attention modules to segmentation performance, and evaluating our network on real volumes to assess its practical utility. Our experimental results demonstrate that the proposed Co-ERA-Net achieves superior segmentation performance compared to existing state-of-the-art networks, with both the co-supervision and enhanced region attention modules contributing significantly to overall performance. Furthermore, the Co-ERA-Net exhibits strong robustness in evaluating real volumes.

The main contributions of the paper are listed below:We present that lung infections occur only within the lung region. It offers valuable inspiration for developing segmentation methodologies about diverse infections, incorporating lung region information into deep learning algorithms and dataset construction.We propose the new Co-ERA-Net for infection segmentation in the chest images. Current deep learning algorithms primarily focus on whole images, but co-supervision with lung region information from our proposed Co-ERA-Net can help the network better concentrate on high-probability infection areas within the lung region.We also introduce Enhanced Region Attention Module (ERAM) to connect lung region and infection flows for more effective information utilization. Our enhanced region attention fuses information from both lung and infection regions to generate region attention as a hint for the infection area.We carefully conduct a series of experiments to evaluate our models from different perspectives, including comparisons with state-of-the-art models, ablation studies to validate the effects of co-supervision and enhanced region attention, and real volume predictions to verify our model’s robustness in actual medical scenarios.

## 2. Related Work

In the context of deep learning applied to chest image infection segmentation, it is crucial to investigate the COVID-19 infection segmentation in both chest CT scans and chest X-ray scans. In the past, the COVID-19 pandemic presented a significant challenge, necessitating the development of efficient infection segmentation methods. Although COVID-19 has subsided considerably, the knowledge and techniques acquired can be utilized to create a tool for infection segmentation that can be extended to other types of infections. This section offers an overview of existing works on COVID-19 infection segmentation and investigates mechanisms that can be generalized to accommodate other infection types.

### 2.1. Deep Learning for COVID-19 Infection Segmentation: Progress and Challenges

Deep learning has emerged as a powerful tool for medical diagnosis, particularly during the COVID-19 pandemic. Using deep learning algorithms to identify infected regions in chest CT and X-ray scans presents a promising strategy for reducing radiologists’ workload and improving diagnostic efficiency. As a result, numerous studies have focused on deep learning-based approaches for infection segmentation, with the analysis of the challenges in accurate segmentation of infection [1].

However, the limited availability of training data hinders the application of deep learning for chest CT and X-ray infection segmentation. To address this issue, multiple datasets and benchmarks have been proposed for training and validating deep learning-based infection segmentation algorithms. In the CT scans, Ma et al. [2]. provided a COVID-19 infection segmentation dataset consisting of 20 public COVID-19 CT scans, while MedSeg [3] offers another dataset containing 9 axial, volumetric CTs. These datasets facilitate the development and validation of deep learning-based COVID-19 chest CT infection segmentation algorithms. In the Chest X-ray scans, Degerli et al. [4] proposes the COVID-19 infection segmentation Chest X-ray dataset consisting of 2951 COVID-19 samples with their corresponding infection masks, while Tahir et al.  [5] provided another COVID-19 classification and segmentation dataset consisting of 11,956 COVID-19 samples with their ground-truth lung masks, which is based on the dataset proposed by Degerli et al. [4]. Furthermore, Ma et al. [6]. proposed the first evaluation benchmark for COVID-19 infection segmentation, which includes three tasks for lung and infection segmentation based on 70 annotated COVID-19 cases, along with baseline models for comparison. With the increasing number of COVID-19 infection segmentation datasets available, various deep learning networks have been developed, such as the pioneer deep learning network for COVID-19 infection segmentation by Fan et al. [7] in CT images, which uses reverse attention to incorporate edge information and improve segmentation accuracy, and the Chest X-ray infection map generation network proposed by Degerli et al. [4], which proposes a novel method for the joint localization, severity grading, and detection of COVID-19 from Chest X-ray images. More details of the state-of-the-art COVID-19 infection segmentation model are shown in Table 1. In addition to localizing COVID-19 infection, Li et al. [8] introduced a deep-learning-based pipeline for directly assessing the four clinical stages of COVID-19 using CT images. This pioneering approach establishes a benchmark for computer-aided diagnosis of COVID-19 by incorporating infection segmentation into the assessment process.

Despite rapid progress, current networks primarily focus on entire CT or X-ray slices rather than lung regions, which have a higher likelihood of containing the infection. This approach may result in inaccurate segmentation outside lung regions and overlooked infections within them. Some recent studies attempt to address this issue by incorporating ground-truth lung masks [12] or lung region segmentation models [14]. However, these approaches introduce challenges such as the unavailability of ground-truth lung masks for new cases and the increased time cost of sequential models. To overcome these limitations, we propose a novel Co-supervision mechanism that simultaneously supervises network training using both infection and lung region masks and an Enhanced Region Attention mechanism that augments lung region information to regions with a high probability of containing the infection. This approach leverages lung region information while eliminating the need for ground-truth lung masks or external lung segmentation models in new cases.

### 2.2. Co-Supervision from Multiple Targets

Utilizing multiple targets to provide supplementary information is a common strategy for enhancing deep learning network performance when single-target supervised learning reaches its limit. In infection segmentation networks, the second supervision target is often the edge information of the primary target. For instance, Fan et al. [7] proposed an infection segmentation network that uses both the infection mask’s edge information and the mask itself, capturing multiple infection perspectives to improve segmentation accuracy. Hu et al. [11] presented a deep collaborative supervision scheme that employs the Edge Supervised Module and Auxiliary Semantic Supervised Module to guide the network in learning edge features and semantics in infection regions, integrating information at each scale through the Attention Fusion Module.

Co-supervision, combining the primary target and its edge information, can enhance infection segmentation performance. However, solely relying on edge information may not provide reliable indicators of the target’s location. This is primarily due to the inherent susceptibility of edge information to noise and low contrast, making it challenging for co-supervision to discern weak edges that are typically noisy and have low contrast, a common scenario in radiology images [15]. This can lead to inconsistent segmentation results. However, region-specific information exhibits higher saliency compared to edge information. In image segmentation, saliency refers to the distinct quality of an object, pixel, or individual that allows it to stand out from its surroundings. For accurate infection region segmentation, it is more reliable to use highly salient region-specific information than weak edge information. Hence, by acknowledging the fact that lung-related infections are primarily found within the lung region, we adopted lung region information instead of edge information for co-supervision. This modification offers more potent and salient indications, improving the network’s efficiency in identifying infection regions.

### 2.3. Attention Mechanism

Attention mechanisms emulate human cognitive processes that selectively focus on specific objects while disregarding others, leading to improved target observation. Initially applied to neural machine translation [16], attention mechanisms were later extended to natural language processing by the Transformer [17], which employed self-attention to capture relationships between words and sentences, enhancing network comprehension.

Due to its effectiveness, attention mechanisms have been adapted in various ways for computer vision. Xu et al. [18]. demonstrated the distinction between soft and hard attention and applied soft attention to the Image Caption task, resulting in superior performance. Woo et al. [19]. proposed channel attention and spatial attention, integrating them into convolution blocks for performance improvement. Task-specific attention mechanisms have also been developed, such as Shen et al. [20]’s region attention, which leverages semantic and edge information for target object region decisions. While region attention can quickly aid infection segmentation tasks, similar textures and boundaries of infected regions may reduce attention efficiency. Consequently, enhancing region attention using lung region masks and infection information is essential for guiding the network to accurately predict infection regions.

### 2.4. Addressing Limitations: An Analysis of Gaps in Current Works

Existing literature on chest infection segmentation using deep learning reveals some gaps and limitations that need to be addressed. Predominantly, most of the works are focused on complete radiological image analyses rather than specifically targeting the lung regions where infections are more likely to be displayed. This approach of complete radiological image analyses can erroneously lead to the segmentation of areas outside the lung, thus compromising the accuracy of diagnosis. To improve precision, the current models either utilize ground-truth lung masks, limiting their applicability due to potential non-availability in new cases, or employ separate models for lung and infection segmentation, which increase time costs and do not fully leverage the shared features between these two segmentation processes.

Furthermore, co-supervision with target edge information, though effective in certain scenarios, may not always provide a reliable indication of the target location, hinting at the need for lung region information for co-supervision. Another gap is identified in the currently used attention mechanisms; region attention, despite its usefulness, may be less efficient in infection segmentation tasks due to the similar textures and boundaries of infected regions, leading to a requirement for enhanced region attention using lung region masks and infection information for better accuracy.

These gaps underline the need for a novel model that offers an efficient, precise solution by targeting specific lung regions, leveraging lung region information for co-supervision, and implementing an enhanced region attention for more accurate predictions in infection segmentation tasks.

## 3. Method

In this section, we unveil the architecture of our proposed network. We delve into the details of the Enhanced Region Attention module and the loss function we’ve introduced.

### 3.1. Proposed Co-ERA-Net: Co-Supervised Infection Segmentation Utilizing Enhanced Region Attention

This section presents a novel deep-learning network called Co-ERA-Net, specifically designed for infection segmentation, leveraging Co-Supervision and Enhanced Region Attention to ensure exceptional precision. We provide an exhaustive elucidation of the network architecture, the mechanisms of Co-Supervision, the concept of Enhanced Region Attention, and the distinct loss function employed. The network proposed herein mitigates the limitations of dependency on lung region information alone by incorporating both lung region and infection data for Co-Supervision during the training phase. This approach significantly enhances the identification of infection regions. The Co-supervision methodology merge infection and lung region data at all scales, efficiently identifying the impactful areas of lung images and offering more refined locational information as compared to the exclusive use of infection region edge data. To render this approach viable, we implement multi-scale supervision for both lung region and infection flows, thereby enabling multi-scale Co-supervision.

Additionally, we introduce the Enhanced Region Attention Module (ERAM) that incorporates infection region data to refine region attention obtained from lung region data, resulting in enhanced precision and segmentation performance. The detailed workflow of Co-Supervision and ERAM could be seen in Algorithm 1 and Figure 2.
**Algorithm 1** Co-Supervision Scheme and Enhanced Region Attention Working Steps**Input:** Input Slice X, Ground-Truth Infection Mask $M_{inf}$, Ground-Truth Lung Mask $M_{lung}$**Output:** Predicted Infection Mask $Pred_{inf}$, Predicted Lung Mask $Pred_{Lung}$1:Extract the general features *F* from input slice using feature extractor;2:Concurrently, feed *F* into both lung region flow and lung infection flow. Denote *F* as $F^{0}$;3:**for** each level of layer i **do**4:    The lung region flow generates $F_{Lung}^{i}$ from $F^{i-1}$;5:    The lung infection flow generates $F_{inf}^{i}$ from $F^{i-1}$;6:    The Enhanced Region Attention Module accepts $F_{Lung}^{i}$ and $F_{inf}^{i}$, producing the attentive features $F_{Att}^{i}$;7:    Generate the lung region mask for multi-scale supervision $M_{lung}^{i}$ from $F^{\left(i-1\right)}$;8:    Generate the lung infection mask for multi-scale supervision $M_{inf}^{i}$ from $F^{\left(i-1\right)}$;9:    The next level of lung infection flow receives $F_{inf}$ and $F_{Att}$ as input, while the next level of the lung region flow takes in $F_{lung}^{i}$;10:    **return** The lung region mask from the final level $F_{Lung}^{4}$, The lung infection mask from the final level $F_{Inf}^{i}$

We also propose a custom hybrid loss function, combining binary cross-entropy loss, Structural Similarity (SSIM) loss, and Intersection Over Union (IoU) loss to balance accuracy and robustness. As illustrated in Figure 3, our network architecture processes input slices with a feature extractor, generating general features. These features are fed into both the lung region and lung infection flows concurrently. Within both flows, features are decoded at different scales and passed through layers connected with enhanced region attention to refine infection information. The lung region mask and infection mask are generated at the outputs of the corresponding flows. To ensure training stability, we employ multi-scale supervision in both flows, which is shown in Figure 4.

### 3.2. Enhanced Region Attention Module (ERAM): Refining Region Attention

Our proposed Enhanced Region Attention Module (ERAM), depicted in Figure 5, refines the region attention generated from the lung region flow. The lung region flow offers coarse region guidance for the entire lung region and is combined with information from the infection flow to produce attentive infection features. However, this coarse region attention is insufficient in accurately highlighting the infection region. To address this limitation, we introduce a refinement region attention mechanism, consisting of a pyramid structure of convolutional layers with varying dilation rates to extract detailed information from different receptive fields.

The refined region attention identifies regions with a high probability of infection and is fused with the information from the infection flow before being used as input for the subsequent stage of the lung infection flow. Our Enhanced Region Attention Module (ERAM) improves segmentation accuracy by refining the attention generated from the lung region flow, effectively leveraging information from both the lung region and infection flows.

### 3.3. Loss Function: Supervising Infection Segmentation in CT Scans

In this work, we propose a loss function composed of two components, Lung Region Loss and Lung Infection Loss, designed to optimize our network for infection segmentation in chest images. The Lung Region Loss component is a Binary Cross-Entropy Loss that supervises the probability distribution of regions used in attention. The Lung Infection Loss component is a hybrid loss that combines Binary Cross-Entropy Loss, Intersection Over Union Loss, and Structural Similarity Measurement Loss. The Binary Cross-Entropy Loss term supervises the probability distributions between the predicted infection and ground truth, while the Intersection Over Union Loss term supervises the overlap between the predicted infection mask and ground truth. Lastly, the Structural Similarity Measurement Loss term supervises the structural similarity between the predicted and ground truth infection masks. The specific formulas for the loss function are provided below:(1)$$BCELoss\left(i\right)=-[y_{i}\times logx_{i}+\left(1-y_{i}\right)\times log\left(1-x_{i}\right)]$$
(2)$$IoULoss\left(i\right)=1-\frac{\sum (x_{i}\times y_{i})}{\sum (x_{i}+y_{i}-x_{i}\times y_{i})}$$
(3)$$SSIMLoss\left(i\right)=1-\frac{\left(2\times \mu_{x_{i}}\times \mu_{y_{i}}\right)\times \left(2\times \sigma_{x_{y}}\right)}{\left(\mu_{x_{i}}^{2}+\mu_{y_{i}}^{2}\right)\times \left(\sigma_{x_{i}}^{2}+\sigma_{y_{i}}^{2}\right)}$$
where $x_{i}$ is the predicted result, $y_{i}$ is the ground-truth, $\mu_{x_{i}}$ and $\mu_{y_{i}}$ are the pixel sample mean, ${\sigma_{x_{i}}}^{2}$ and ${\sigma_{y_{i}}}^{2}$ are the variance and $\sigma_{xy}$ is the covariance.

To enable multi-scale supervision in the lung region and lung infection flows, we apply the same loss function across all scales but assign varying weights to each component. This approach effectively balances the relative importance of different loss functions across scales, enhancing overall training stability and accuracy.

## 4. Experimental Results and Discussion

### 4.1. Dataset Description: Utilizing Publicly Available COVID-19 Segmentation Datasets in CT and Chest X-ray

The study utilizes three publicly accessible datasets for network training and testing. These datasets are as follows: the COVID-19 CT Lung and Infection Segmentation Dataset (CT-LISD), the Segmentation nr.2 Dataset (CT-S2D), and the COVID-QU-Ex Dataset (X-ray-QUED). Detailed information about the datasets is provided in Table 2.

CT-LISD encompasses 20 COVID-19 CT scans with 3520 slices in total. For the sake of training stability, 1844 slices containing infections were selected for network training. CT-S2D contains nine axial volumetric CTs from Radiopaedia, comprising 829 slices. When assessing segmentation, only 373 infection-containing slices were utilized. However, for real CT volume evaluation, all 829 slices were used to ensure a comprehensive performance assessment. X-ray-QUED, sourced from various medical centers, holds 5826 slices. Following the dataset’s official split and segmentation evaluation using only infection-containing slices, we assigned 1864 slices for network training, 583 slices for testing, and 466 slices for validation. It should be noted that all datasets provide ground truth lung masks, critical for the Co-Supervision aspect of our network design.

### 4.2. Implementation Details: Network Training and Configuration

The Co-ERA-Net, our proposed solution, was implemented using PyTorch (Version 1.7.1). The feature extraction was carried out using ConvNext, pretrained on ImageNet for improved feature extraction capabilities. We selected the AdamW optimizer with an initial learning rate of 0.0002. The weights of loss functions in each level are correspondingly 0.2, 0.3, 0.4 and 1.

The input data were resized to dimensions of 256 × 256, with the data loader’s batch size set at four. Data augmentation was exclusively done using the RandomFlip method, with a probability of 0.5, and no other augmentation strategies were adopted. Training of the network was performed on an NVIDIA RTX 3080 GPU with 12 GB memory, and it took place over 200 epochs, with the convergence and fine-tuning time approximately being 20 h. The training loss curves for training our Co-ERA-Net are shown in Figure 6.

To ensure a fair and equal comparison of performance between our model and other state-of-the-art models in the experiments, we retrained all the networks using their official implementations. This retraining was done on the same training dataset as used for our network, with identical training settings, including input size, batch size, optimizer, and number of epochs. By adhering to consistent training conditions, we aimed to eliminate any potential bias and ensure a reliable evaluation of the models’ comparative performance. we also conduct a comprehensive analysis of various network parameters, Floating Point Operations per Second (FLOPs), training time, and inferencing time. The statistical results are presented in Table 3.

### 4.3. Evaluation Metrics: Assessing Infection Segmentation Performance

To evaluate the performance of our experiments in infection segmentation, we used four metrics: Intersection over Union(IoU), Dice Coefficient, Mean Absolute Error, and Saliency F-Score. The Intersection over Union and Dice Coefficient are widely used metrics in medical image segmentation, measuring the similarity between the generated infection mask and the ground-truth infection mask. The Mean Absolute Error is a pixel-level metric that quantifies the accuracy of infection segmentation. The Saliency F-Score is a metric that combines precision and recall to evaluate the accuracy of binary infection segmentation. We report both Maximum and Mean types of Saliency F-Score to provide a comprehensive assessment.

### 4.4. Quantitative Evaluation: Comparing with State-of-the-Art Models

We evaluated our proposed Co-ERA-Net against a variety of state-of-the-art models, including U-Net family, general segmentation networks, medical segmentation networks, and other infection segmentation networks, specifically for the infection segmentation of CT datasets. The comparative results are provided in Table 4. Our Co-ERA-Net surpassed all the other models across every evaluation metric, displaying a consistently robust performance. This improvement is primarily due to the integration of the Co-supervision scheme, lung region information, and our Enhanced Region Attention Module. The lung region information directs the network to focus on the most effective areas for infection segmentation. In addition, our Enhanced Region Attention Module further refines the lung region information, enabling accurate identification of regions with high infection probability.

We also extended our evaluation to the Chest X-ray dataset. We conducted a similar comparative experiment using the same models as those in the CT datasets. Table 5 presents the results of infection segmentation from the test set of the Chest X-ray dataset, and Table 6 shows the results from its validation set. The Co-ERA-Net maintained superior performance compared to other models across all evaluation metrics for the Chest X-ray datasets. This further emphasizes the effectiveness of the lung region information and the Enhanced Region Attention Module in improving infection segmentation performance beyond CT images.

### 4.5. Qualitative Evaluation: Visualizing Prediction Results

Our proposed network demonstrates superior infection segmentation in CT and Chest X-ray images, as depicted in Figure 7 and Figure 8, outperforming other baseline networks. Notably, our network effectively avoids mis-segmenting infection regions and closely aligns with ground-truth masks.

This superior performance is due to the integration of lung region information and the Enhanced Region Attention Module. These factors focus the network on relevant areas and refine segmentation by emphasizing high-probability regions, leading to significant improvements in infection segmentation.

Despite challenges in Chest X-ray images, such as noise and interference, our network maintains precision. It achieves accurate segmentation by effectively identifying infected areas and refining segmentation through larger receptive fields. As a result, our network achieves precise infection segmentation in Chest X-ray images, outperforming other baseline networks.

### 4.6. Ablation Study: Examining Key Components

This section presents an ablation study assessing the contributions of three critical elements in our proposed network: lung region information, enhanced region attention, and a custom hybrid loss function with multi-scale supervision.

Firstly, we analyzed the impact of integrating lung region information on infection segmentation accuracy in chest radiology images. A comparison of two baselines, one with and one without lung region information, showed a considerable accuracy improvement with the inclusion of lung region information, as shown in Table 7 for CT images and Table 8 and Table 9 for Chest X-ray images. Figure 9 (CT images) and Figure 10 (Chest X-ray images) visually reinforce these benefits, highlighting reduced redundant infection segmentation when lung region information is incorporated.

Next, we introduced the Enhanced Region Attention Module to address the limitations of lung region information alone. By comparing two models— one with lung region information and the other with both lung region information and enhanced region attention— we identified a significant performance boost with the addition of the Enhanced Region Attention Module (Table 7 for CT images and Table 8 and Table 9 for Chest X-ray images). Figure 9 (CT images) and Figure 10 (Chest X-ray images) also demonstrate that solely using lung region information can lead to mis-segmentation, whereas the Enhanced Region Attention Module effectively employs lung region information to pinpoint infection regions.

We also conducted a comprehensive ablation study to compare our proposed Enhanced Region Attention Module with several other attention mechanisms in the field of deep learning. These attention mechanisms fall into two categories: those integrated with the convolution block, such as the Convolution Block Attention Module [19], Squeeze-Excitation Attention [33], and Triplet Attention [34], and those working independently, including Pyramid Attention [35], Parallel Reverse Attention [36], and Multi-scale Self-Guided Attention [37]. To assess their performance, we conducted experiments on both CT images (Table 7) and Chest X-ray images (Table 8 and Table 9). The results demonstrates that our proposed Enhanced Region Attention outperforms the other attention mechanisms. Notably, visual inspection of the segmentation results using our attention module (Figure 11 for CT images and Figure 12 for Chest X-ray images) reveals its remarkable ability to avoid mis-segmentation within lung regions, leading to highly accurate segmentation outcomes. The superiority of our Enhanced Region Attention can be attributed to its remarkable capacity to emphasize lung regions and effectively highlight areas with a high probability of infection. Unlike other attention mechanisms that rely on indirect information, such as channel/spatial attention utilization in CBAM and Triplet Attention Mechanism, or reverse attention in Parallel Reverse Attention, our Enhanced Region Attention takes a more direct path to identify infection regions, resulting in significantly higher efficiency. Furthermore, compared to attention mechanisms with similar structures, such as multi-scale supervision and pyramid attention, our Enhanced Region Attention benefits from its ability to focus on specific lung region information, further enhancing the efficiency of these mechanisms. While it’s worth mentioning that our proposed Enhanced Region Attention requires ground-truth lung region masks during the training process, unlike some other attention mechanisms, it demonstrates a stronger capability to utilize information and achieves higher accuracy in COVID-19 infection segmentation.

Finally, we evaluated the contribution of the custom hybrid loss function in enhancing infection mask quality, compared to the commonly used Binary Cross-Entropy Loss function. We observed superior results when combining the custom hybrid loss function with multi-scale supervision, as demonstrated in Table 7 for CT images and Table 8 and Table 9 for Chest X-ray images. Figure 9 (CT images) and Figure 10 (Chest X-ray images) also confirm that the custom hybrid loss function and multi-scale supervision can generate more precise regions of infection and clearer boundaries.

In short, our ablation study accentuates the significant roles of lung region information, enhanced region attention, and the custom hybrid loss with multi-scale supervision in augmenting infection segmentation accuracy in our proposed network.

### 4.7. Evaluating Model Performance on Diverse Volumes

The application of deep learning models to real-life volumes, primarily containing infection-free slices, presents challenges due to potential mis-segmentations and reduced model robustness. This section assesses the ability of our proposed model to prevent mis-segmentation in infection-free slices.

Our model was first tested on both infected (373 slices) and non-infected (456 slices) CT slices, despite having been trained only on infected slices. Slices without infection severity information, i.e., with blank infection masks, were considered infection-free. We defined true positives, true negatives, false positives, and false negatives as follows: a true positive is when the model correctly segments infection in infected slices; a false negative occurs when the model misses segmentation in infected slices; a false positive happens when the model erroneously segments infection in infection-free slices; a true negative is when the model rightly excludes infection in infection-free slices. Based on these definitions, we calculated the confusion matrix and performance metrics. Similar tests were applied to both infected (1049 slices) and non-infected (1049 slices) Chest X-ray slices to further evaluate our model’s effectiveness.

Figure 13 presents the confusion matrix, and Table 10 reveals the results of discriminating between infected and infection-free CT and Chest X-ray slices. Assisted by lung region information and enhanced region attention, our model effectively detected infection in CT and Chest X-ray slices, even when trained only on infected slices, underscoring the real-world applicability and robustness of our proposed Co-ERA-Net for volumes comprising both infected and non-infected slices. Figure 14 displays continuous predictions for entire CT volumes, demonstrating our network’s proficiency in handling infection-free slices while maintaining stable predictions across the volumes. Due to the unavailability of X-ray volumes in the Chest X-ray dataset, we couldn’t provide the continuous prediction for Chest X-ray volumes.

## 5. Conclusions

This paper presents the Co-ERA-Net, a novel segmentation network leveraging lung region information for enhanced COVID-19 chest infection segmentation in CT and X-ray images. It highlights the crucial role of lung region information in accurate lesion segmentation, a challenge encountered in the realm of COVID-19 image analysis. Our proposed scheme outperforms existing methodologies, affirming its efficacy and utility.

Our proposed methodology stands out for its novelty in deep learning, leveraging lung region information in the context of COVID-19 imaging. Firstly, we introduce the Co-supervision scheme, which assimilates lung region information at multiple scales into the network’s decoding stage, effectively guiding feature extraction and enhancing overall performance. Secondly, the use of enhanced region attention allows us to refine lung region information, resulting in significantly improved segmentation accuracy. Lastly, our network demonstrates remarkable robustness in avoiding mis-segmentation in infection-free slices, showcasing its real-world applicability and reliability. In conclusion, the Co-ERA-Net, with its high efficiency in segmenting infections and its ability to avoid mis-segmentation in infection-free slices, significantly improves the efficiency of daily radiology practice in computer-aided diagnosis. This advancement enables radiologists to focus their attention on slices with infections and prioritize their efforts for patients requiring urgent care, ultimately streamlining and enhancing the diagnostic process for better patient outcomes.

Although our Co-ERA-Net is highly efficient in segmenting the infection inside the COVID-19 radiology images, it still could not avoid having two limitations. First of all, the network’s high efficiency in segmenting infection within COVID-19 radiology images is commendable, but the inability to provide the severity estimation is a notable limitation. Addressing this challenge would require collecting more extensive COVID-19 data, and carefully classifying detailed infection categories. Furthermore, the observation of sub-optimal learning performance when applying the model to X-ray scans compared to CT scans raises an important consideration for future work. The discrepancy might arise from the network being based on CT images while X-ray scans correspond to CT scans in their reduced dimensional forms [38]. To overcome this issue, the future direction of designing a network to transform X-ray images into CT-like representations before applying the previously trained CT infection segmentation network demonstrates a possible solution toward making the framework more universal and effective across different imaging modalities.

## Figures and Tables

**Figure 1 bioengineering-10-00928-f001:**
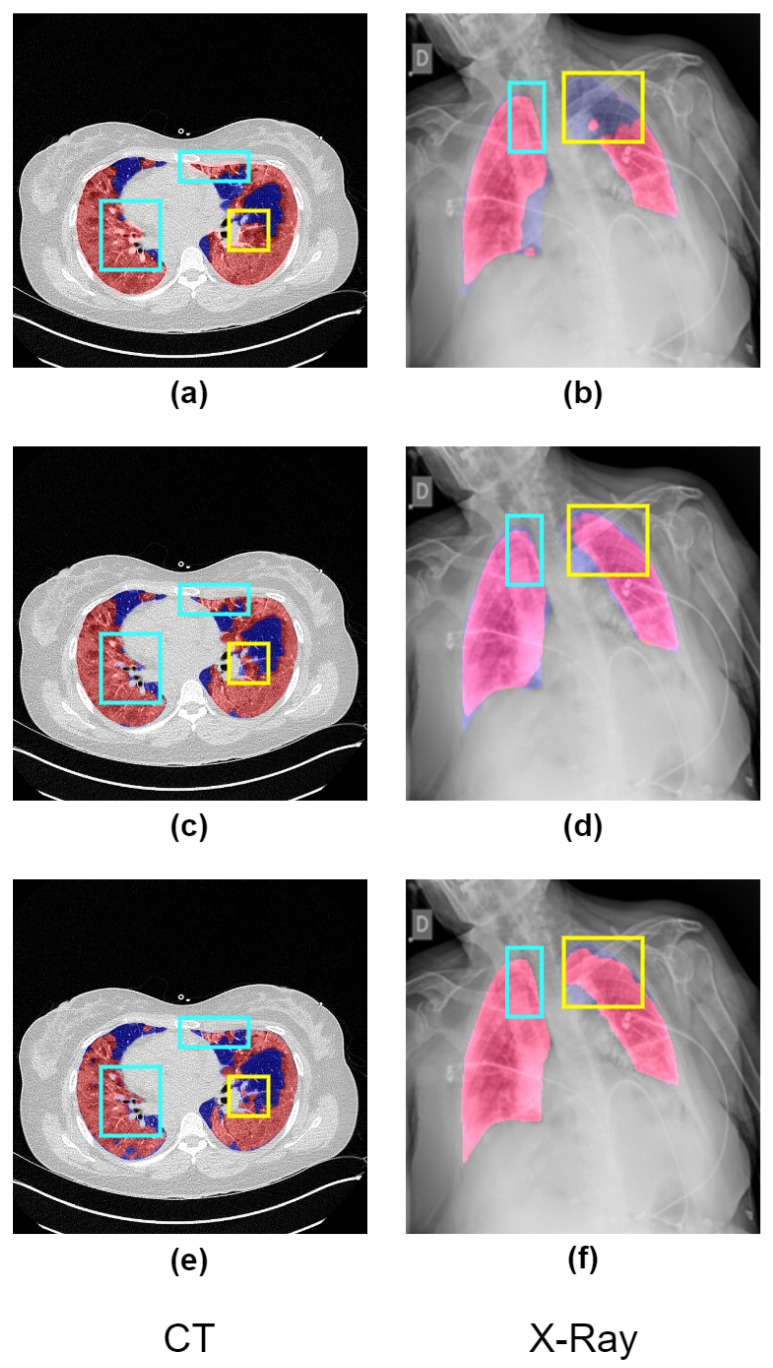
The graphic presents a visual comparison of the effectiveness of different methods in identifying infected regions (highlighted in red) in COVID-19 CT and X-ray images. The ground-truth lung regions (highlighted in blue) provided by datasets are annotated by the senior radiologists. In images (**a**,**b**), the segmentation results of a deep learning model lacking prior knowledge of lung regions are illustrated. It is evident from these images that there are misidentified regions outside the lung area (shown in blue) and inaccuracies in segmentation (indicated by yellow). Our proposed solution, Co-ERA-Net, is demonstrated in images (**c**,**d**). These images clearly show how our solution effectively mitigates the issues of misidentification and inaccurate segmentation seen in (**a**,**b**). Finally, images (**e**,**f**) serve as the ground truth against which these different methods are compared. They provide the benchmark to evaluate the performance of the deep learning model and the Co-ERA-Net solution.

**Figure 2 bioengineering-10-00928-f002:**
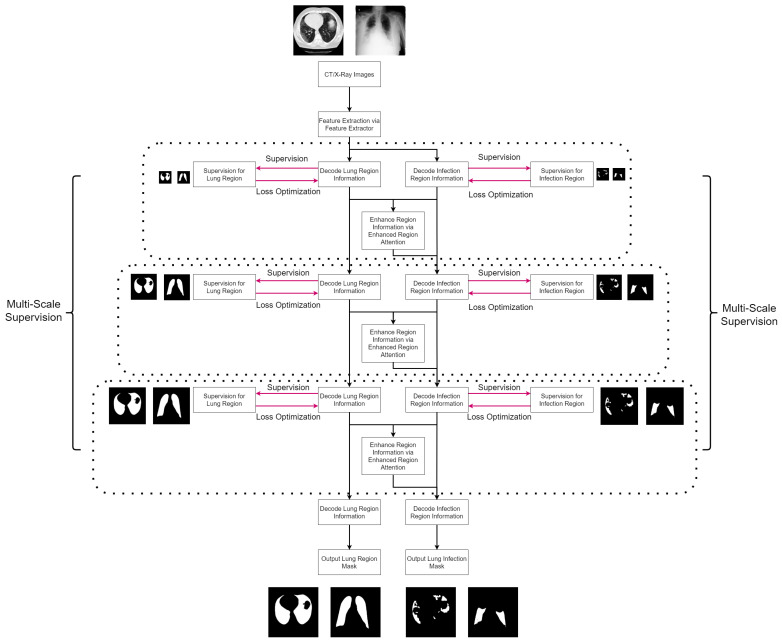
The proposed algorithm flowchart for lung infection segmentation. Initially, the input image undergoes feature extraction to obtain general features. These general features are then simultaneously utilized in both the lung region stream and the lung infection stream, producing corresponding lung region features and lung infection features in parallel. To identify regions with a high probability of infection, the lung region features and lung infection features are subjected to the enhanced region attention mechanism, generating attentive features. The attentive features and lung infection features are combined and forwarded to the next level of the lung infection stream, enhancing the algorithm’s ability to capture significant patterns related to infections. To facilitate stable and effective training, multi-scale supervision is applied to both the lung region stream and lung infection stream, enabling the algorithm to learn essential features at various scales and improving overall accuracy in lung infection segmentation.

**Figure 3 bioengineering-10-00928-f003:**
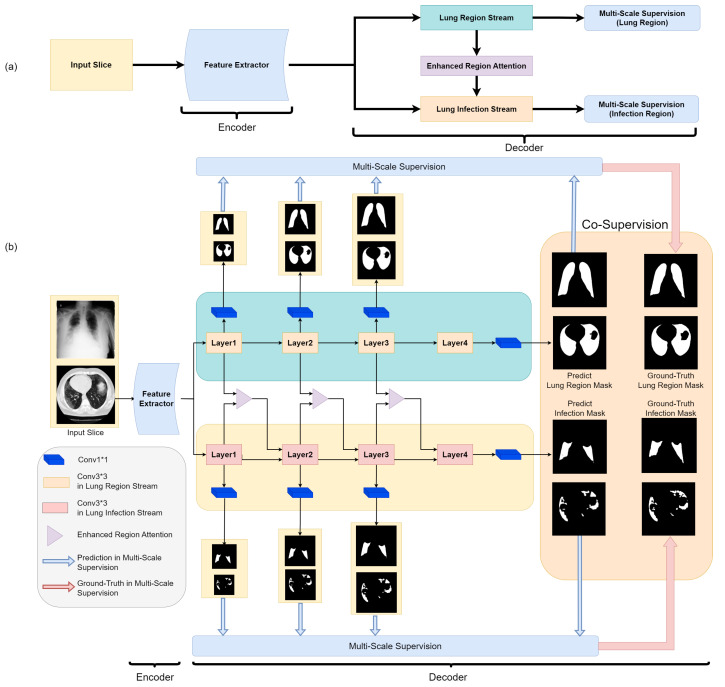
The block diagram of our proposed Co-ERA-Net (**a**) and the detailed Illustration of the overall network architecture (**b**). The proposed architecture comprises of Co-Supervision scheme and Enhanced Region Attention Module based on the encoder-decoder structure. The Co-Supervision scheme is used to highlight the effective area of infection. The Enhanced Region Attention Module is employed to strengthen the region information by integrating the infection information into the lung region information for high-probability region information.

**Figure 4 bioengineering-10-00928-f004:**
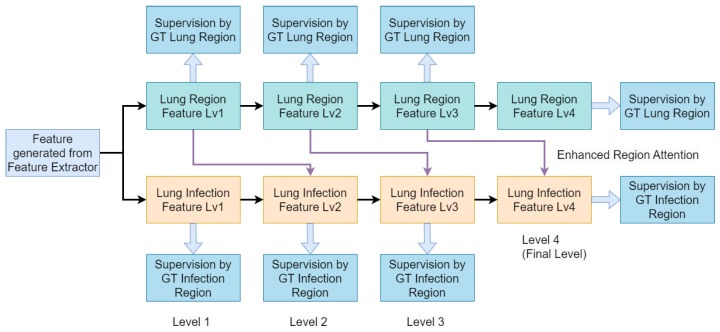
Block diagram of our Multi-Scale Co-Supervision Scheme. Our proposed scheme incorporates lung region information and infection information as supervision targets at each scale of the decoder. Additionally, we employ enhanced region attention as the connection between the lung region feature and lung infection feature to reinforce infection feature representation.

**Figure 5 bioengineering-10-00928-f005:**
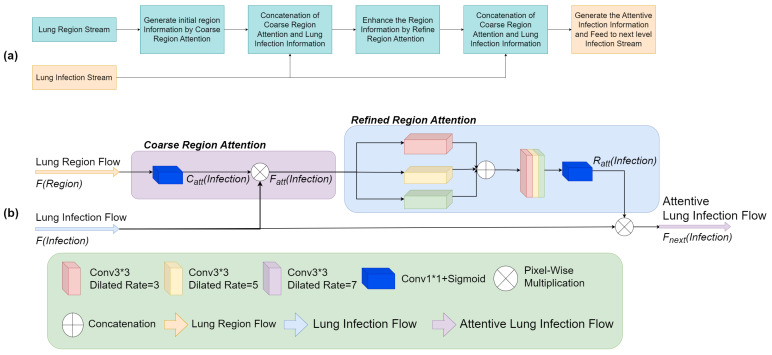
The block diagram of our proposed Enhanced Region Attention Module (**a**) and the detailed illustration of the mechanism (**b**). F(Region) represents the information from the lung region flow and the F(Infection) represents the information from the infection flow. The F(Region) is generated the coarse region attention $C_{att}\left(Infection\right)$ in the Coarse Region Attention stage and integrated with the F(Infection) as the attentive infection information $F_{att}\left(Infection\right)$. The $F_{att}\left(Infection\right)$ is duplicated and fed into the Refined Region Attention stage which is constructed by the convolution layer with different dilated rates to generate the refined attention $R_{att}\left(Infection\right)$. The $R_{att}\left(Infection\right)$ and F(Infection) are fused and generate the $F_{next}\left(Infection\right)$ as the input of the next level infection flow.

**Figure 6 bioengineering-10-00928-f006:**
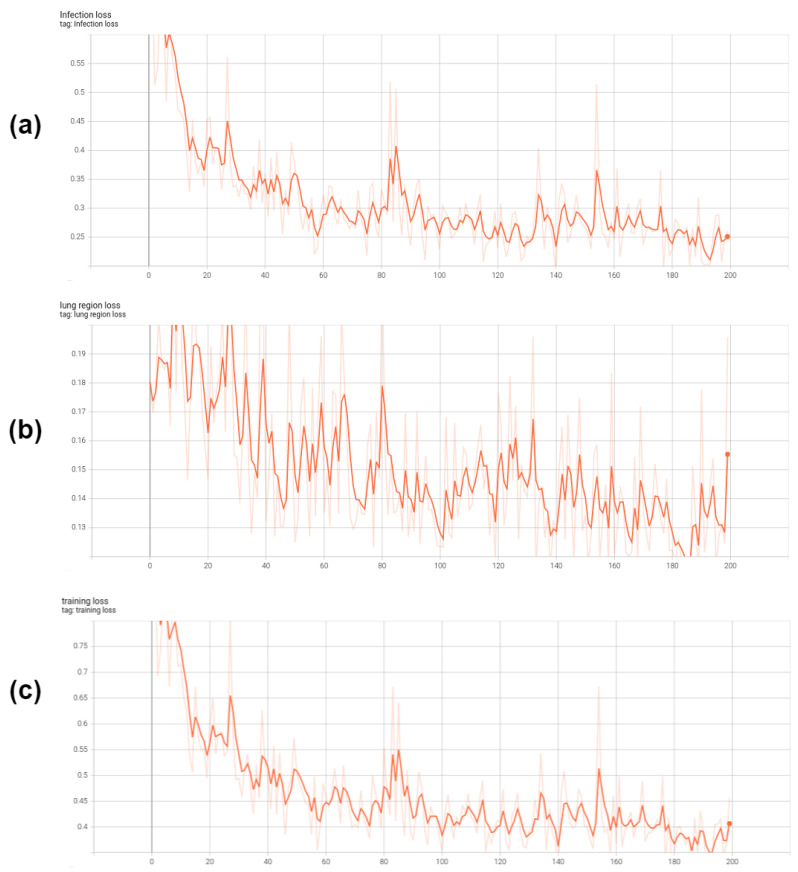
The training loss curves of our Co-ERA-Net. (**a**) is the infection loss, (**b**) is the lung region loss and (**c**) is the total loss which is the combination of lung region loss and infection loss.

**Figure 7 bioengineering-10-00928-f007:**
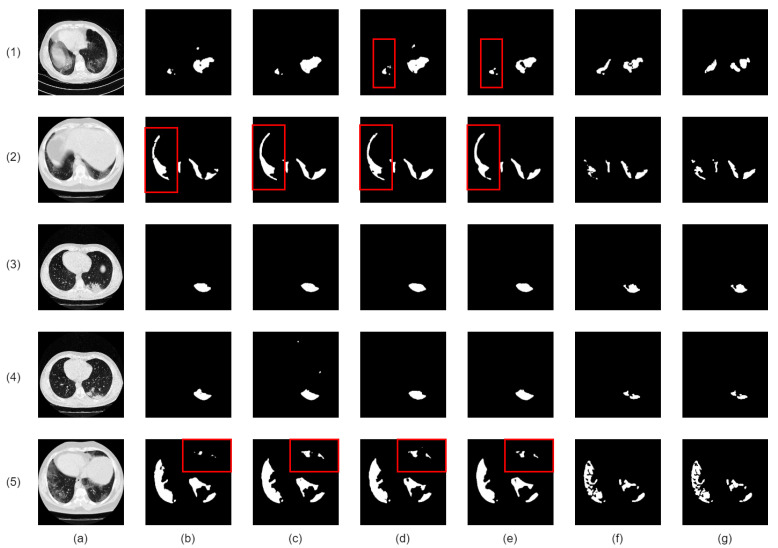
Visual qualitative comparison of lung infection segmentation result among Attention UNet, SegFormer, nnUNet, CoupleNet and our proposed method from CT images. (**a**): Input CT slice; (**b**): Attention UNet; (**c**): SegFormer; (**d**): CPFNet; (**e**): CoupleNet; (**f**): Our Co-ERA-Net; (**g**): The corresponding ground truth (GT). Red boxes highlight the wrong segmentation.

**Figure 8 bioengineering-10-00928-f008:**
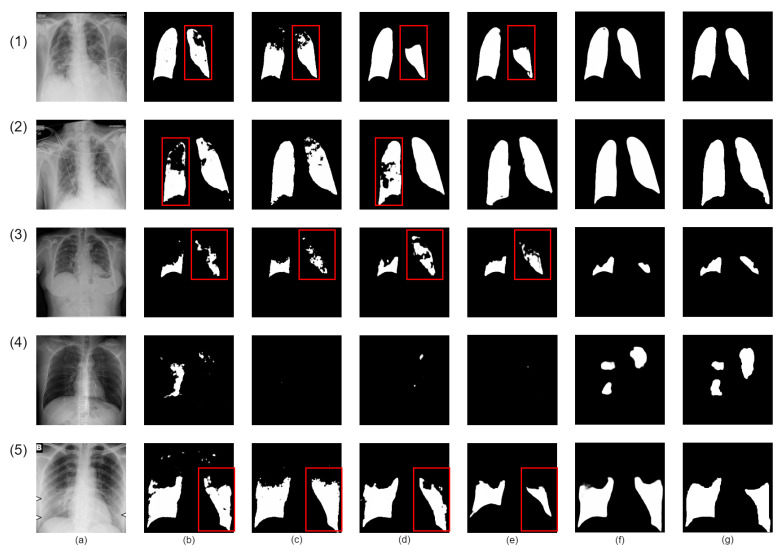
Visual qualitative comparison of lung infection segmentation result among Attention UNet, SegFormer, nnUNet, CoupleNet and our proposed method from X-ray images. (**a**): Input X-ray slice; (**b**): Attention UNet; (**c**): SegFormer; (**d**): CPFNet; (**e**): BSNet; (**f**): Our Co-ERA-Net; (**g**): The corresponding ground truth (GT). Red boxes highlight the wrong segmentation.

**Figure 9 bioengineering-10-00928-f009:**
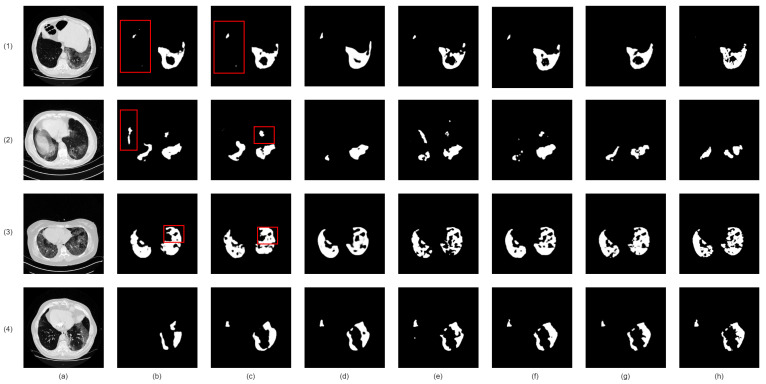
Visual comparison of lung infection segmentation result from CT images in Ablation Study. (**a**): Input CT slice; (**b**): Co-ERA-Net without lung region information and enhanced region attention; (**c**): Co-ERA-Net without enhanced region attention; (**d**): Co-ERA-Net trained by single-scale Binary Cross-Entropy Loss; (**e**): Co-ERA-Net trained by single-scale proposed hybrid loss; (**f**): Co-ERA-Net trained by multi-scale Binary Cross-Entropy Loss; (**g**): Co-ERA-Net trained by multi-scale proposed hybrid loss; (**h**): the corresponding ground truth (GT). Red boxes highlight the wrong segmentation.

**Figure 10 bioengineering-10-00928-f010:**
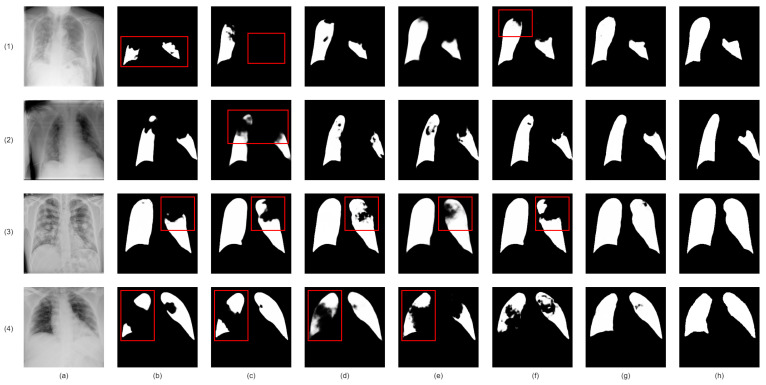
Visual comparison of lung infection segmentation result from chest X-ray images in Ablation Study. (**a**): Input X-ray slice; (**b**): Co-ERA-Net without lung region information and enhanced region attention; (**c**): Co-ERA-Net without enhanced region attention; (**d**): Co-ERA-Net trained by single-scale Binary Cross-Entropy Loss; (**e**): Co-ERA-Net trained by single-scale proposed hybrid loss; (**f**): Co-ERA-Net trained by multi-scale Binary Cross-Entropy Loss; (**g**): Co-ERA-Net trained by multi-scale proposed hybrid loss; (**h**): the corresponding ground truth (GT). Red boxes highlight the wrong segmentation.

**Figure 11 bioengineering-10-00928-f011:**
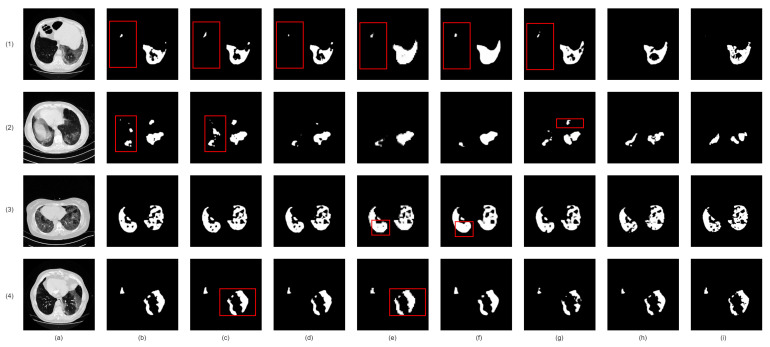
Visual comparison of lung infection segmentation results with other attention mechanisms from CT images in Ablation Study. (**a**): Input CT slice; (**b**): Convolution Block Attention Module; (**c**): Squeeze-Excitation Attention; (**d**): Triplet Attention; (**e**): Pyramid Attention; (**f**): Parallel Reverse Attention; (**g**): Multi-scale Self-Guided Attention; (**h**): Co-ERA-Net with Enhanced Region Attention; (**i**): the corresponding ground truth (GT). Red boxes highlight the wrong segmentation.

**Figure 12 bioengineering-10-00928-f012:**
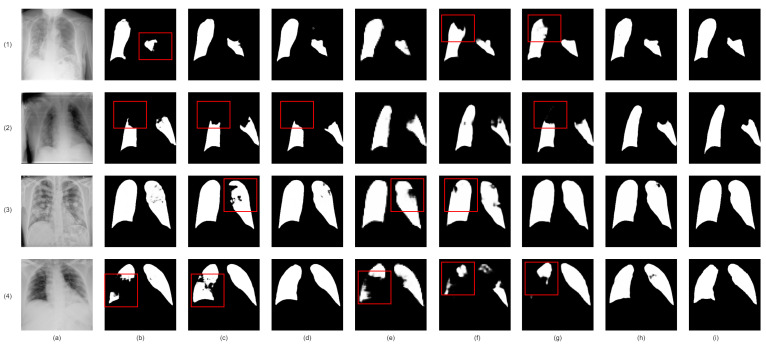
Visual comparison of lung infection segmentation results with other attention mechanisms from chest X-ray images in Ablation Study. (**a**): Input CT slice; (**b**): Convolution Block Attention Module; (**c**): Squeeze-Excitation Attention; (**d**): Triplet Attention; (**e**): Pyramid Attention; (**f**): Parallel Reverse Attention; (**g**): Multi-scale Self-Guided Attention; (**h**): Co-ERA-Net with Enhanced Region Attention; (**i**): the corresponding ground truth (GT). Red boxes highlight the wrong segmentation.

**Figure 13 bioengineering-10-00928-f013:**
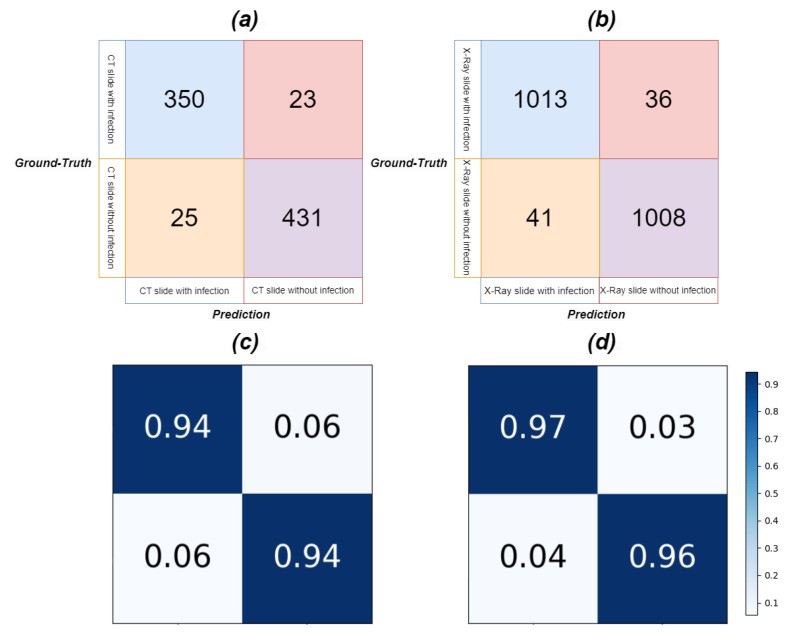
The Confusion Matrix of slices with and without infection for CT and X-ray images in the evaluation of our model performance on diverse volumes. (**a**) is the confusion matrix for CT, (**b**) is the confusion matrix for X-ray, (**c**) is the normalized confusion matrix for CT and (**d**) is the normalized confusion matrix for X-ray.

**Figure 14 bioengineering-10-00928-f014:**
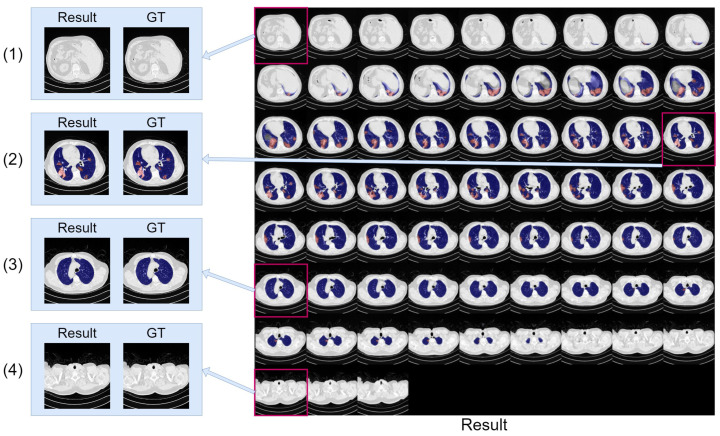
The montage of CT volume slices showing the segmentation results generated by our model. The infection region is highlighted in red, and the lung region is highlighted in blue for clarity. The situations containing both lung region and infection (**2**), only lung region (**3**), and no lung region and infection region (**1**,**4**) are enlarged and compared with ground truth (GT).

**Table 1 bioengineering-10-00928-t001:** The pros and cons of the State-of-the-art COVID-19 infection segmentation method.

Author	Network	Features	Pros	Cons
Fan et al. [7]	Inf-Net	• Aggregating features from high-level layers using a parallel partial decoder (PPD) • Recurrent reverse attention (RA) modules • Edge-attention guidance • Semi-supervised learning strategy.	• Boundary identification based on reverse attention and edge constraint guidance. • Semi-supervised learning to overcome the shortage of labeled data.	• Accuracy drop for non-infected slices. • Two-step strategy for multi-class labeling resulted in sub-optimal learning performance.
Wang et al. [9]	COPLE-Net	• Noise-robust loss • Adaptive self-ensembling framework with exponential moving average (EMA)	• Noise-robust dice loss Function to handle noisy annotations during the training process. • Exponential moving average (EMA) teacher model to guide a standard student model, enhancing the model’s robustness against noisy labels.	• The performance under a wider range of noise levels have not been explored yet. • It lacks an investigation into the network’s performance under various noise levels
Qiu et al. [10]	MiniSeg	• Attentive hierarchical spatial pyramid • Extremely minimum network	• Attentive hierarchical spatial pyramid improves the representation capabilities in lightweight multi-scale learning. • Extremely minimum network complexity makes it suitable for practical implementation in resource-constrained scenarios.	• Extremely low complexity of the network limited the generalizability through different datasets.
Hu et al. [11]	Deep collaborative supervision network	• Deep collaborative supervision scheme • Auxiliary semantic supervised module • Attention fusion module • Edge supervised module	• Deep collaborative supervision scheme enhances supervised information of different levels and fuses different scale features maps. • Edge spervised module allows the model to capture rich spatial information at various scales.	• Limited severity estimation. It is not sufficient for estimating the severity of infected COVID-19. • Simultaneously applying the co-supervision scheme in both the down-sampling and up-sampling paths leads to decreased segmentation performance.
Paluru et al. [12]	AnamNet	• Apply lung extraction before infection segmentation • Fully convolutional anamorphic depth block • Adapted label weighting scheme	• Apply lung extraction before infection segmentation enhances the efficiency to find out infection. • Fully convolutional anamorphic depth block enabled efficient gradient flow in the network.	• The network’s limitation lies in its restriction to 2D chest CT images. • It shows inherent bias towards the peripheral part of the lung in its segmentation.
Cong et al. [13]	BSNet	• End-to-end boundary guided semantic learning • Dual-branch semantic enhancement • Mirror-symmetric boundary guidance	• Boundary-guided semantic learning leverages boundary guidance and semantic relations to capture infection areas. • Dual-branch semantic enhancement model semantic relations, enhancing the feature learning process. • Mirror-symmetric boundary guidance ensuring complementary and sufficiency of feature learning.	• It faces difficulty in segmenting COVID-19 infections due to the scattered nature of infected regions over the chest slice.

**Table 2 bioengineering-10-00928-t002:** Datasets used in the experiments.

Author	Dataset	Imaging Modality	Image Bits	Number of Total Slices	Number of Slices with Infection	Data Split
Ma et al. [2]	Zenodo 20P	CT	8 bit	3520	1844	Train: 1844
MedSeg et al. [3]	Radiopaedia 9P	CT	8 bit	829	373	Test: 373
						Train: 1864
Tahir et al. [5]	COVID-QU-Ex Dataset	X-ray	8 bit	5826	2913	Test: 583
						Validation: 466

**Table 3 bioengineering-10-00928-t003:** Network Parameters, Flops, training time for convergence and inference time of the State-of-the-art models and our model in the experiments.

Network	Network Parameters	FLOPs	Training Time (Hours)	Inference Time (Seconds)
**UNet Family**				
Attention UNet [21]	9.16 M	34.86 G	17.81	0.1628
UNet [22]	39.39 M	80.45 G	22.99	0.2118
UNet++ [23]	47.17 M	199.69 G	43.11	0.5949
UNet+++ [24]	26.97 M	199.67 G	44.12	0.7044
**General Segmentation Network in Natural **				
FCN [25]	18.64 M	25.51 G	9.54	0.0880
Deeplab V3 [26]	59.33 M	22.18 G	8.74	0.0798
PSPNet [27]	49.06 M	48.56 G	16.64	0.1528
SegFormer [28]	7.71 M	20.15 G	20.48	0.1863
**Medical Segmentation Network **				
Double UNet [29]	97.58 M	211.78 G	35.94	0.4263
CE-Net [30]	29.00 M	8.89 G	5.51	0.0503
CPFNet [31]	30.65 M	8.03 G	7.85	0.0727
Medical-Transformer [32]	1.37 M	2.40 G	56.50	0.7166
**State-of-the-Art Infection Segmentation Network **				
Inf-Net [7]	31.07 M	7.36 G	9.40	0.0853
COPLE-Net [9]	10.52 M	11.18 G	6.80	0.0626
MiniSeg [10]	0.08 M	0.12 G	3.00	0.0274
Deep Collaborative Supervision Network [11]	29.18 M	48.94 G	2.82	0.0261
AnamNet [12]	4.63 M	25.402 G	2.54	0.0232
BSNet [13]	43.98 M	45.75 G	1.69	0.0156
**Co-ERA-Net (Ours) **	70.37 M	20.49 G	1.35	0.0125

**Table 4 bioengineering-10-00928-t004:** Performance comparisons between different networks in lung infection segmentation from CT images. The red text represents the best results.

Network	IoU↑		Dice↑		MAE ↓		F-Score↑		
	Mean	STD	Mean	STD	Mean	STD	Max	Mean	STD
**UNet Family**									
Attention UNet [21]	0.5598	0.1794	0.7456	0.1628	0.0071	0.0076	0.7653	0.7059	0.1640
UNet [22]	0.5634	0.1712	0.7401	0.1608	0.0074	0.0080	0.7601	0.7023	0.1644
UNet++ [23]	0.5455	0.1744	0.7334	0.1663	0.0074	0.0072	0.7578	0.6837	0.1692
UNet+++ [24]	0.5612	0.1755	0.7407	0.1638	0.0072	0.0073	0.7712	0.6939	0.1592
**General Segmentation Network in Natural **									
FCN [25]	0.5457	0.1740	0.7358	0.1582	0.0075	0.0076	0.7581	0.6832	0.1532
Deeplab V3 [26]	0.5425	0.1726	0.7364	0.1668	0.0073	0.0076	0.7688	0.6903	0.1680
PSPNet [27]	0.5418	0.1711	0.7330	0.1650	0.0076	0.0078	0.7488	0.6858	0.1720
SegFormer [28]	0.5568	0.1739	0.7380	0.1636	0.0074	0.0074	0.7447	0.6860	0.1711
**Medical Segmentation Network **									
Double UNet [29]	0.5464	0.1839	0.7200	0.1842	0.0075	0.0070	0.7137	0.6655	0.1915
CE-Net [30]	0.5581	0.1769	0.7379	0.1665	0.0072	0.0072	0.7422	0.6870	0.1737
CPFNet [31]	0.5656	0.1743	0.7420	0.1655	0.0071	0.0073	0.7658	0.6951	0.1606
Medical-Transformer [32]	0.5807	0.1702	0.7348	0.1529	0.0076	0.0081	0.7613	0.7030	0.1719
**State-of-the-Art Infection Segmentation Network **									
Inf-Net [7]	0.5201	0.1754	0.7244	0.1740	0.0079	0.0078	0.7339	0.6662	0.1790
COPLE-Net [9]	0.5630	0.1636	0.7522	0.1554	0.0067	0.0066	0.7724	0.7077	0.1545
MiniSeg [10]	0.5240	0.1919	0.6967	0.1699	0.0080	0.0079	0.7568	0.6862	0.1570
Deep Collaborative Supervision Network [11]	0.5992	0.1809	0.6759	0.1546	0.0079	0.0080	0.7725	0.6872	0.1545
AnamNet [12]	0.5668	0.1723	0.7506	0.1605	0.0068	0.0069	0.7706	0.7082	0.1738
BSNet [13]	0.5391	0.1726	0.7302	0.1681	0.0076	0.0077	0.7497	0.6766	0.1792
**Co-ERA-Net (Ours) **	0.6553	0.1517	0.7945	0.1435	0.0050	0.0054	0.8373	0.7984	0.1325

**Table 5 bioengineering-10-00928-t005:** Performance comparisons between different networks in lung infection segmentation from the test set of chest X-ray images. The red text represents the best results.

Network	IoU↑		Dice↑		MAE ↓		F-Score↑		
	Mean	STD	Mean	STD	Mean	STD	Max	Mean	STD
**UNet Family**									
Attention UNet [21]	0.6192	0.2307	0.7453	0.2309	0.0499	0.0341	0.7737	0.7299	0.2193
UNet [22]	0.5758	0.2183	0.7004	0.1989	0.0588	0.0364	0.7424	0.7078	0.2157
UNet++ [23]	0.6276	0.2229	0.7377	0.1997	0.0489	0.0344	0.7684	0.7437	0.2115
UNet+++ [24]	0.6299	0.2235	0.7377	0.1932	0.0496	0.0333	0.7738	0.7419	0.2092
**General Segmentation Network in Natural **									
FCN [25]	0.6515	0.2328	0.7541	0.2082	0.0461	0.0368	0.7851	0.7611	0.2181
Deeplab V3 [26]	0.6144	0.2335	0.7260	0.2057	0.0529	0.0362	0.7719	0.7248	0.2230
PSPNet [27]	0.6338	0.2294	0.7440	0.1988	0.0487	0.0370	0.7902	0.7453	0.2136
SegFormer [28]	0.5962	0.2243	0.7046	0.2033	0.0586	0.0430	0.7621	0.7216	0.2167
**Medical Segmentation Network **									
Double UNet [29]	0.6581	0.2352	0.7673	0.2093	0.0439	0.0350	0.7771	0.7571	0.2246
CE-Net [30]	0.6538	0.2372	0.7566	0.2111	0.0455	0.0352	0.7743	0.7526	0.2245
CPFNet [31]	0.6579	0.2398	0.7547	0.2159	0.0424	0.0331	0.7913	0.7617	0.2215
Medical Transformer [32]	0.5828	0.2506	0.6994	0.2377	0.0565	0.0431	0.7049	0.7021	0.2426
**State-of-the-Art Infection Segmentation Network **									
Inf-Net [7]	0.6451	0.2365	0.7516	0.2399	0.0418	0.0386	0.7942	0.7630	0.2368
COPLE-Net [9]	0.6544	0.2308	0.7628	0.2123	0.0423	0.0354	0.7935	0.7632	0.2148
MiniSeg [10]	0.5841	0.2344	0.7092	0.2231	0.0568	0.0386	0.7592	0.7079	0.2353
Deep Collaborative Supervision Network [11]	0.6329	0.2368	0.6456	0.1984	0.0656	0.0434	0.7809	0.6692	0.1967
AnamNet [12]	0.6282	0.2311	0.7358	0.2109	0.0495	0.0348	0.7637	0.7386	0.2230
BSNet [13]	0.6683	0.2370	0.7652	0.2163	0.0421	0.0363	0.7879	0.7680	0.2219
**Co-ERA-Net (Ours) **	0.6736	0.2317	0.7711	0.2094	0.0411	0.0336	0.7989	0.7683	0.2144

**Table 6 bioengineering-10-00928-t006:** Performance comparisons between different networks in lung infection segmentation from the validation set of chest X-ray images. The red text represents the best results.

Network	IoU↑		Dice↑		MAE ↓		F-Score↑		
	Mean	STD	Mean	STD	Mean	STD	Max	Mean	STD
**UNet Family**									
Attention UNet [21]	0.6436	0.2212	0.7687	0.1876	0.0457	0.0334	0.7940	0.7530	0.2040
UNet [22]	0.5948	0.2089	0.7153	0.1863	0.0557	0.0321	0.7583	0.7250	0.2013
UNet++ [23]	0.6499	0.2149	0.7570	0.1852	0.0451	0.0304	0.7863	0.7636	0.1920
UNet+++ [24]	0.6528	0.2099	0.7583	0.1740	0.0472	0.0305	0.7918	0.7617	0.1902
**General Segmentation Network in Natural **									
FCN [25]	0.6665	0.2323	0.7650	0.2108	0.0427	0.0335	0.7910	0.7696	0.2039
Deeplab V3 [26]	0.6234	0.2291	0.7388	0.1937	0.0517	0.0349	0.7817	0.7324	0.2122
PSPNet [27]	0.6549	0.2200	0.7656	0.1859	0.0455	0.0339	0.8027	0.7633	0.1997
SegFormer [28]	0.6160	0.2205	0.7200	0.2020	0.0547	0.0390	0.7706	0.7341	0.2128
**Medical Segmentation Network **									
Double UNet [29]	0.6713	0.2276	0.7808	0.2014	0.0416	0.0354	0.7891	0.7686	0.2143
CE-Net [30]	0.6694	0.2281	0.7701	0.2021	0.0425	0.0334	0.7888	0.7652	0.2125
CPFNet [31]	0.6719	0.2352	0.7635	0.2137	0.0406	0.0317	0.8025	0.7755	0.2095
Medical Transformer [32]	0.5925	0.2466	0.7090	0.2303	0.0563	0.0472	0.7128	0.7100	0.2298
**State-of-the-Art Infection Segmentation Network **									
Inf-Net [7]	0.6574	0.2368	0.7595	0.2462	0.0401	0.0301	0.8060	0.7676	0.2400
COPLE-Net [9]	0.6662	0.2233	0.7711	0.2066	0.0404	0.0294	0.8015	0.7723	0.1902
MiniSeg [10]	0.5960	0.2226	0.7195	0.2032	0.0554	0.0369	0.7712	0.7197	0.2154
Deep Collaborative Supervision Network [11]	0.6472	0.2389	0.7162	0.2238	0.0501	0.0348	0.7903	0.7439	0.2018
AnamNet [12]	0.6420	0.2230	0.7484	0.1995	0.0475	0.0319	0.7760	0.7511	0.2093
BSNet [13]	0.6710	0.2304	0.7779	0.2073	0.0404	0.0277	0.8050	0.7602	0.1868
**Co-ERA-Net (Ours) **	0.6893	0.2329	0.7823	0.2170	0.0389	0.0308	0.8085	0.7756	0.2186

**Table 7 bioengineering-10-00928-t007:** Ablation experiments in the lung infection segmentation from CT images. The red text represents the best results.

Network	Lung Region	Enhanced Region Attention		IoU↑		Dice↑		MAE ↓		F-Score↑		
				Mean	STD	Mean	STD	Mean	STD	Max	Mean	STD
Baseline				0.5094	0.1686	0.6521	0.1815	0.0099	0.0117	0.6866	0.6500	0.1796
Baseline	✔			0.5512	0.1783	0.7262	0.1751	0.0077	0.0080	0.7485	0.6862	0.7485
Co-ERA-Net	✔	✔		0.6553	0.1517	0.7945	0.1435	0.0050	0.0054	0.8373	0.7984	0.1325
Network	Multi-Scale	BCE Loss	Hybird Loss	IoU↑		Dice↑		MAE ↓		F-Score↑		
				Mean	STD	Mean	STD	Mean	STD	Max	Mean	STD
Co-ERA-Net		✔		0.5096	0.1728	0.7211	0.1696	0.0078	0.0077	0.7589	0.6732	0.1797
Co-ERA-Net			✔	0.5665	0.1729	0.7471	0.1622	0.0071	0.0073	0.7650	0.7020	0.1748
Co-ERA-Net	✔	✔		0.6011	0.1794	0.7481	0.1723	0.0073	0.0086	0.7708	0.7196	0.1794
Co-ERA-Net	✔		✔	0.6553	0.1517	0.7945	0.1435	0.0050	0.0054	0.8373	0.7984	0.1325
Network	Lung Region	Attention Type	Work Type	IoU↑		Dice↑		MAE ↓		F-Score↑		
				Mean	STD	Mean	STD	Mean	STD	Max	Mean	STD
Baseline	✔	Convolution Block Attention Module [19]	With Blocks	0.5550	0.1745	0.7401	0.1673	0.0070	0.0070	0.7643	0.6959	0.1699
Baseline	✔	Squeeze-Excitation Attention [33]	With Blocks	0.5615	0.1765	0.7400	0.1670	0.0070	0.0070	0.7615	0.6974	0.1708
Baseline	✔	Triplet Attention [34]	With Blocks	0.5618	0.1768	0.7417	0.1651	0.0070	0.0069	0.7641	0.7004	0.1697
Baseline	✔	Pyramid Attention [35]	Independent	0.4768	0.1588	0.7045	0.1645	0.0085	0.0086	0.7429	0.6556	0.1653
Baseline	✔	Parallel Reverse Attention [36]	Independent	0.4901	0.1681	0.7178	0.1682	0.0082	0.0082	0.7547	0.6638	0.1664
Baseline	✔	Multi-scale Self-Guided Attention [37]	Independent	0.5583	0.1788	0.7395	0.1668	0.0069	0.0072	0.7749	0.7097	0.1617
Co-ERA-Net	✔	Enhanced Region Attention	Independent	0.6553	0.1517	0.7945	0.1435	0.0050	0.0054	0.8373	0.7984	0.1325

**Table 8 bioengineering-10-00928-t008:** Ablation experiments in the lung infection segmentation from the test set of Chest X-ray images. The red text represents the best results.

Network	Lung Region	Enhanced Region Attention		IoU↑		Dice↑		MAE ↓		F-Score↑		
				Mean	STD	Mean	STD	Mean	STD	Max	Mean	STD
Baseline				0.6446	0.2431	0.7488	0.2267	0.0428	0.0358	0.7833	0.7594	0.2297
Baseline	✔			0.6640	0.2417	0.7660	0.2095	0.0412	0.0336	0.7890	0.7643	0.2143
Co-ERA-Net	✔	✔		0.6736	0.2317	0.7711	0.2094	0.0411	0.0336	0.7989	0.7683	0.2144
Network	Multi-Scale	BCE Loss	Hybird Loss	IoU↑		Dice↑		MAE ↓		F-Score↑		
				Mean	STD	Mean	STD	Mean	STD	Max	Mean	STD
Co-ERA-Net		✔		0.6442	0.2317	0.7519	0.2045	0.0440	0.0362	0.7941	0.7532	0.2149
Co-ERA-Net			✔	0.6680	0.2360	0.7676	0.2119	0.0413	0.0375	0.7973	0.7590	0.2078
Co-ERA-Net	✔	✔		0.6695	0.2330	0.7661	0.2034	0.0415	0.0353	0.7952	0.7514	0.2115
Co-ERA-Net	✔		✔	0.6736	0.2317	0.7711	0.2094	0.0411	0.0336	0.7989	0.7683	0.2144
Network	Lung Region	Attention Type	Work Type	IoU↑		Dice↑		MAE ↓		F-Score↑		
				Mean	STD	Mean	STD	Mean	STD	Max	Mean	STD
Baseline	✔	Convolution Block Attention Module [19]	With Blocks	0.6479	0.2512	0.7431	0.2366	0.0434	0.0383	0.7905	0.7568	0.2135
Baseline	✔	Squeeze-Excitation Attention [33]	With Blocks	0.6551	0.2450	0.7548	0.2212	0.0436	0.0382	0.7910	0.7629	0.2155
Baseline	✔	Triplet Attention [34]	With Blocks	0.6570	0.2414	0.7564	0.2160	0.0432	0.0360	0.7911	0.7604	0.2083
Baseline	✔	Pyramid Attention [35]	Independent	0.6097	0.2276	0.7355	0.2178	0.0488	0.0345	0.7848	0.7344	0.2045
Baseline	✔	Parallel Reverse Attention [36]	Independent	0.6349	0.2255	0.7343	0.2137	0.0461	0.0371	0.7914	0.7640	0.1882
Baseline	✔	Multi-scale Self-Guided Attention [37]	Independent	0.6193	0.2208	0.7370	0.2114	0.0483	0.0377	0.7918	0.7483	0.1939
Co-ERA-Net	✔	Enhanced Region Attention	Independent	0.6736	0.2317	0.7711	0.2094	0.0411	0.0336	0.7989	0.7683	0.2144

**Table 9 bioengineering-10-00928-t009:** Ablation experiments in the lung infection segmentation from the validation set of Chest X-ray images. The red text represents the best results.

Network	Lung Region	Enhanced Region Attention		IoU↑		Dice↑		MAE ↓		F-Score↑		
				Mean	STD	Mean	STD	Mean	STD	Max	Mean	STD
Baseline				0.6721	0.2417	0.7566	0.2265	0.0414	0.0302	0.7955	0.7713	0.2307
Baseline	✔			0.6742	0.2450	0.7657	0.2283	0.0398	0.0316	0.8048	0.7717	0.2275
Co-ERA-Net	✔	✔		0.6893	0.2329	0.7823	0.2170	0.0389	0.0308	0.8085	0.7756	0.2186
Network	Multi-Scale	BCE Loss	Hybird Loss	IoU↑		Dice↑		MAE ↓		F-Score↑		
				Mean	STD	Mean	STD	Mean	STD	Max	Mean	STD
Co-ERA-Net		✔		0.6305	0.2249	0.7509	0.2121	0.0433	0.0324	0.7865	0.7594	0.2092
Co-ERA-Net			✔	0.6843	0.2422	0.7761	0.2241	0.0398	0.0306	0.7991	0.7680	0.2246
Co-ERA-Net	✔	✔		0.6773	0.2307	0.7719	0.2097	0.0392	0.0291	0.7917	0.7654	0.2117
Co-ERA-Net	✔		✔	0.6893	0.2329	0.7823	0.2170	0.0389	0.0308	0.8085	0.7756	0.2186
Network	Lung Region	Attention Type	Work Type	IoU↑		Dice↑		MAE ↓		F-Score↑		
				Mean	STD	Mean	STD	Mean	STD	Max	Mean	STD
Baseline	✔	Convolution Block Attention Module [19]	With Blocks	0.6711	0.2459	0.7603	0.2308	0.0405	0.0342	0.7944	0.7606	0.2048
Baseline	✔	Squeeze-Excitation Attention [33]	With Blocks	0.6710	0.2394	0.7666	0.2156	0.0405	0.0328	0.7925	0.7650	0.2040
Baseline	✔	Triplet Attention [34]	With Blocks	0.6767	0.2354	0.7729	0.2151	0.0408	0.0347	0.7997	0.7674	0.1990
Baseline	✔	Pyramid Attention [35]	Independent	0.6272	0.2178	0.7521	0.2042	0.0457	0.0310	0.7903	0.7511	0.1830
Baseline	✔	Parallel Reverse Attention [36]	Independent	0.6432	0.2173	0.7435	0.2155	0.0445	0.0318	0.8004	0.7685	0.1795
Baseline	✔	Multi-scale Self-Guided Attention [37]	Independent	0.6222	0.2268	0.7377	0.2238	0.0467	0.0356	0.7872	0.7463	0.2013
Co-ERA-Net	✔	Enhanced Region Attention	Independent	0.6893	0.2329	0.7823	0.2170	0.0389	0.0308	0.8085	0.7756	0.2186

**Table 10 bioengineering-10-00928-t010:** The evaluation metrics of slices with and without infection in our model based on the confusion matrix in Figure 13. We set the negative prediction when the model outputs a blank mask while the positive prediction when the model outputs the mask with infection.

(**a**) CT	
	**Value**
Accuracy	0.9420
Sensitivity	0.9383
Specificity	0.9451
precision	0.9333
F1 score	0.9358
Matthews correlation coefficient	0.8864
(**b**) X-ray	
	**Value**
Accuracy	0.9632
Sensitivity	0.9609
Specificity	0.9656
precision	0.9655
F1 score	0.9632
Matthews correlation coefficient	0.9279

## Data Availability

The data presented in this study are openly available and it could be checked in Table 2.

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
