# Peer review of "Co-ERA-Net: Co-Supervision and Enhanced Region Attention for Accurate Segmentation in COVID-19 Chest Infection Images"

_bioengineering, 2023, doi:10.3390/bioengineering10080928_

Round 1

Reviewer 1 Report

The paper proposes a segmentation method for chest images that aids the detection of image regions where infections appear. The proposed method combines supervised training and multiscale image processing.

The literature is well reviewed.

The paper reads in very good English.

This is a very nice and well written paper. 

All methods are clearly defined in equations and accompanied with elaborate figures.

The experimental conditions are well defined.

Experiments are quantitative and comparative to the state of the art and they are based on publicly available benchmark datasets. An ablation study supports the choices of the critical parameters of the proposed work.

The paper does not exhibit any flaws. Therefore my recommendation is to accept it as is. Nevertheless, there are a few minor comments to provide that target the better presentation of this work.

In Figure 1, can you please mention how the blue (ground truth) regions were found? Were they manually annotated by an expert? 

I enjoyed the references to perceptual tasks, a central component of radiological inspection that is often overlooked. However, in Section 2, authors may wish to investigate the literature a bit further pertinent to the role of saliency in detection tasks and specifically the role of image contrast.

I failed to understand whether 8-bit or 16-bit images are employed in this work.

Author Response

Dear Reviewer 1,

Please see the attachment for our answers to your comments.

Best Regards,

ZeBang He

Reviewer 2 Report

The authors introduce Co-supervision and Enhanced Region Attention Network (Co-ERA-Net) for Accurate Segmentation in COVID-19 Chest Infection Images. The paper is well written but needs some improvements to better understand.

·         You are using Enhanced Region Attention Module (ERAM) in your model, but you did not explain this. The authors are requested to:

­   Explain it clearly with a block diagram.

­   Mention what are the pros and cons of ERAM over other types of attention. 

­   Perform an ablation study with other types of attention.

·         Although Figure 2 illustrates the overall network architecture of the proposed model, the authors should provide a clear explanation of your model to enhance clarity. Moreover, you mentioned that this is an encoder-decoder-based architecture, but it is unclear which part acts as an encoder and decoder.

·         There is missing information regarding multi-scale supervision. The authors are requested to explain it with a proper block diagram.

·         The authors are requested to make a flowchart along with the algorithm to enhance the understanding.

·         Furthermore, it is essential to discuss the limitations of your proposed model.

·         The authors are requested to add the distribution of data used in other SOTA algorithms.

·         Add a table of performance comparison with SOTA algorithms in terms of parameters, complexity, training, and inference time.

·         Add a comparison table of the proposed and state-of-the-art studies highlighting the features and limitations of each model.

·         There is an improvement needed in English writing e.g.,

­   Line 71: We present the assumption that lung infections occur only within the lung.

­   Line 83: We design various experiments to evaluate our models.

­   Line 88: In discussing chest image infection segmentation using deep learning.

Author Response

Dear Reviewer 2,

Please see the attachment for our answers to your comments.

Best Regards,

ZeBang He

Reviewer 3 Report

Authors have presented an interesting method for COVID-19 Segmentation. However, I have the following suggestions to improve the conceptual clarity of the paper further.

1.     The related works section need to be re-written. Many recent papers based on deep learning for COVID detection have been missed. The following CT-based COVID-19 papers need to be included in the related works section to support a healthy discussion in the related works section.

a)     AI for COVID-19 Detection from Radiographs: Incisive Analysis of State of the Art Techniques, Key Challenges and Future Directions

b)     Deep co-supervision and attention fusion strategy for automatic COVID-19 lung infection segmentation on CT images

c)     A deep-learning-based framework for severity assessment of COVID-19 with CT images

2.     The gaps in the existing works related need to be consolidated and discussed at the end of related works section.  

3.     Plots for convergence for loss and accuracy need to be presented.

4.     Limitations and constraints involved need to be discussed.

5.     How the proposed attention is different from the existing methods like CBAM, Triplet attention etc.

6.     Confusion matrix will be suitable to analyze classification results. Instead objective comparison with metrics like IOU, Jaccard can be given to analyze the segmentation performance.

7.     Novelty of the proposed system need to be highlighted properly.

Moderate changes required

Author Response

Dear Reviewer 3,

Please see the attachment for our answers to your comments.

Best Regards,

ZeBang He

Round 2

Reviewer 2 Report

Most of my comments are addressed, I recommend the acceptance of this article in current form.